# Mobile class A β-lactamase gene *bla*GMA-1

**Hirokazu Yano,[1] Masato Suzuki,[1] Lisa Nonaka[2]**

**ABSTRACT**  The first reported *bla*GMA-1 gene encoding gammaproteobacterial mobile class A β-lactamase (GMA-1) was identified in a recently defined mobile DNA element, a strand-biased circularizing integrative element (SE). Increased genomic data revealed the presence of *bla*GMA-1 in marine bacteria, including pathogenic species such as *Vibrio parahaemolyticus* and *Photobacterium damselae*. Herein, we address the substrate range of GMA-1 and how frequently *bla*GMA-1 was acquired by the chromosomes or plasmids via SEs using sequences in a publicly available database. An *Escherichia coli* strain carrying *bla*GMA-1 exhibited resistance to amoxicillin, piperacillin, and carbenicillin, but it remained susceptible to cephalosporins, monobactam, and carbapenems, indicating that GMA-1 belongs to functional group 2c, narrow-spectrum β-lactamases. *bla*GMA-1-flanking sequence analysis for sequences in the RefSeq/GenBank database revealed a total of eight distinct SE-mediated *bla*GMA-1 acquisition events and six SE-independent *bla*GMA-1 acquisition events, including *bla*GMA-1 alone translocation, without involving a specific insertion sequence, or integron. Thus, this study shows that GMA-1 is specialized for penicillin degradation and is mainly disseminated by SEs; however, SE is not the only genetic mechanism transmitting *bla*GMA-1.

**IMPORTANCE**  Despite increasing reports, class A β-lactamases of environmental bacteria remain very poorly characterized, with limited understanding of their transmission patterns. To address this knowledge gap, we focused on a recently designated GMA family β-lactamase gene, *bla*GMA-1, found in marine bacterial genera such as *Vibrio*. This study shows that gammaproteobacterial mobile class A β-lactamase is specialized for penicillin degradation, and *bla*GMA-1 is frequently linked to strand-biased circularizing integrative elements (SEs) in sequences in the RefSeq/GenBank database. Evidence for the implication of SEs in β-lactamase environmental transmission provides insights for future surveillance studies of antimicrobial resistance genes in human clinical settings.

**KEYWORDS**  GMA-1, GMA family, penicillinase, plasmid, SE, *Vibrio*

As the first antibiotics used to treat bacterial infections, β-lactam antibiotics are currently the most consumed antibiotics in many countries (1). However, bacteria can develop β-lactam resistance by overexpression of efflux transporters, inactivation of porins, and acquisition of β-lactamase genes (2, 3). β-Lactamases are organized according to their Ambler class and Bush functional group classifications (4). Several clinically significant β-lactamases fall into class A group 2be (commonly known as extended-spectrum β-lactamases, ESBLs), class A group 2f (carbapenemases), and class B group 3a or 3b metallo-β-lactamases (MBLs) capable of degrading most of the currently available synthetic β-lactams (4). ESBL and MBL genes are often located within or near transposable elements or in class 1 integrons in bacterial genomes. Frequently reported *bla*-mobile DNA element associations include *bla*CTX-M variants with IS*Ecp1* and IS*26* (5), *bla*NDM-1 with IS*CR* elements (6), *bla*IMP-1, and *bla*VIM-1 as gene cassettes with class 1 integron (7, 8).

Address correspondence to Hirokazu Yano, h-yano@niid.go.jp.

The authors declare no conflict of interest.

See the funding table on p. 11.

Class A group 2c β-lactamase families, such as PSE, CARB, and VHH, were identified in *Vibrio* and *Pseudomonas* (9–12). Group 2c members can act on first generation cephalosporins to some extent, but they have very poor activity against second or third generation cephalosporins and carbapenems (9, 11, 12), showing a narrow-spectrum characteristic specialized for penicillin degradation. Several *Vibrio* spp. are fish pathogens, and their possession of class A β-lactamases may have occurred by selection associated with penicillin use for aquaculture.

We previously reported three conjugative multidrug resistance plasmids at an aquaculture site in Japan (13–15). These plasmids, pAQU1(13), pSEA1(14), and pSEA2 (15), carry a gene encoding a class A β-lactamase, named by National Center for Biotechnology Information (NCBI) as gammaproteobacterial mobile class A β-lactamase (GMA-1) (NCBI RefSeq protein ID: WP_012774820.1) (15). GMA-1 shows similarity to group 2c members rather than to ESBL (group 2be) members both over the whole sequence and at motif level (Fig. 1). $bla_{GMA-1}$ is embedded in a recently defined mobile DNA element called strand-biased circularizing integrative element (SE), SE-6945, which is located on a plasmid and chromosome 1 of two *Vibrio alfacsensis* strains (15) (Fig. 2A). *Escherichia coli* transconjugants containing $bla_{GMA-1}$ in SE-6945 exhibited ampicillin resistance (15). $bla_{GMA-1}$ in pAQU1 identified in *Photobacterium damselae* subsp. *damselae* was also found embedded in another SE (16).

SEs commonly have four genes encoding the following products: IntA, tyrosine recombinase; Tfp, tyrosine recombinase fold protein; IntB, large tyrosine recombinase; Srap, SE-associated recombination auxiliary protein; and imperfect inverted repeats, C and C′ at their termini (16) (Fig. 2A). The border regions between SEs and the host genome, including inverted repeats, are called *attL* and *attR* (Fig. 2B) (15). The C and

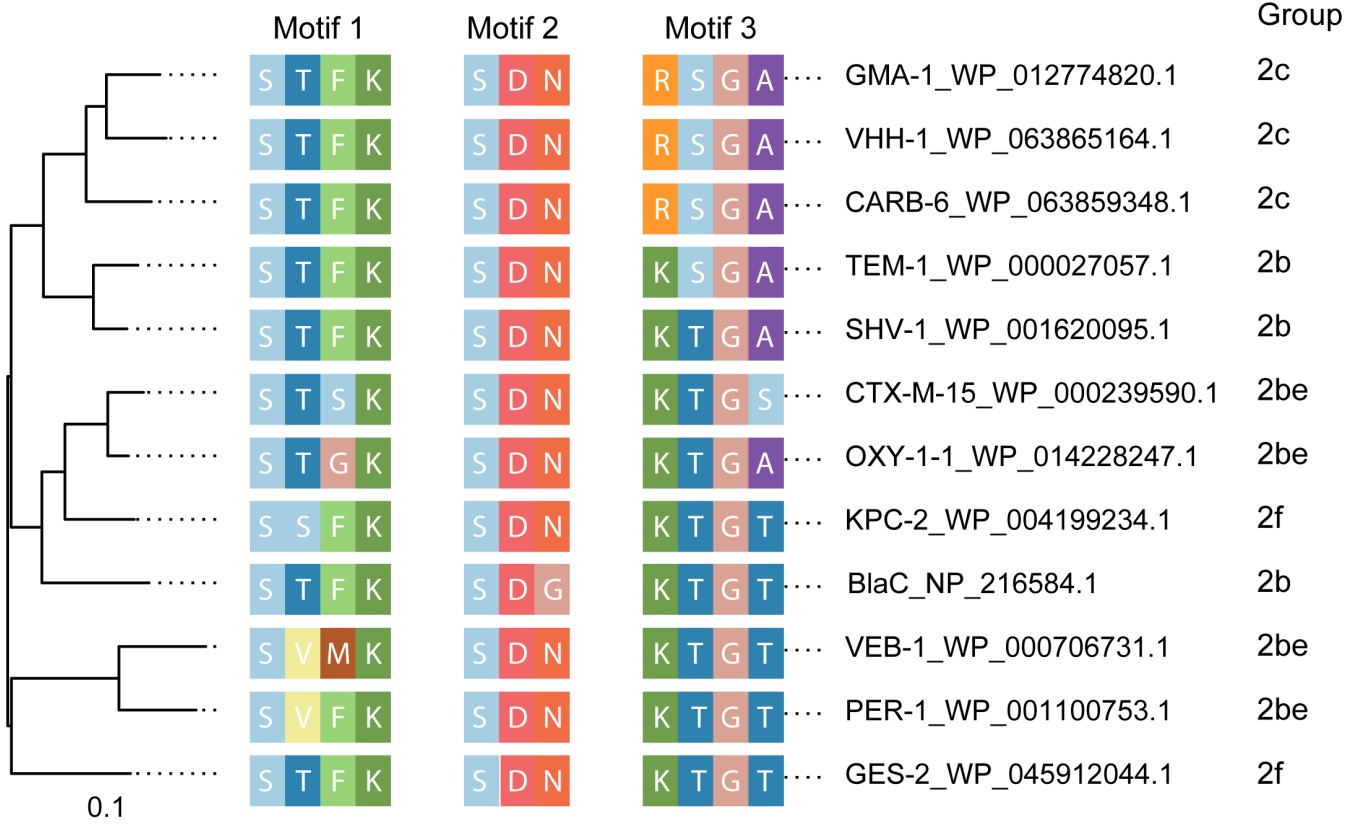

**FIG 1** Phylogenetic tree and functional groups of class A serine β-lactamases. The tree is inferred based on the PROMAL3D (17) alignment of whole amino acid sequences. Three conserved motifs of serine β-lactamases (18) were identified by inspection of the alignment and are shown between tree tips and taxon labels. Functional group assignment is based on Beta-lactamase database (19) and a review (20).

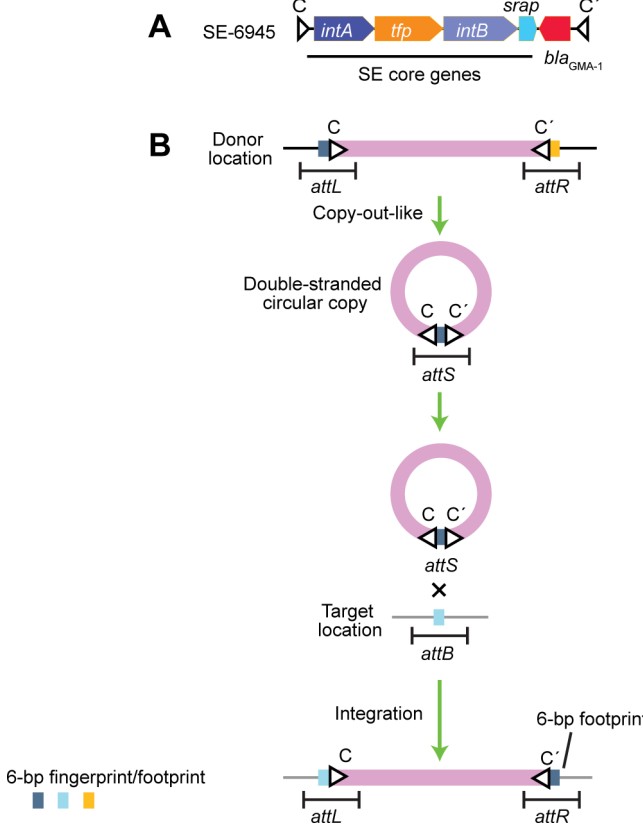

**FIG 2** Genetic organization of SE and the terms used. (A) Genetic organization of SE-6945. Gene names and associated products are as follows: *intA*, tyrosine recombinase; *tfp*, tyrosine recombinase fold protein; *intB*, large tyrosine recombinase; *srap*, SE-associated recombination auxiliary protein. C and C′, imperfect inverted repeats. (B) Rules in the SE movement. The border regions between SE and the host genome, including inverted repeats, are called *attL* and *attR*. The SE generates a double-stranded circular copy containing a 6 bp left flank at the C and C′ joints (*attS*) via a copy-out-like process (16). The target location is called *attB*. The 6 bp in *attS* is placed at the right flank of the new SE copy as a 6 bp footprint after the completion of integration (14, 15). The 6 bp sequence incorporated into *attS* is considered a fingerprint containing information about donor location.

C′ joints on a circular SE copy is termed *attS*, whereas the target location in the host genome is termed *attB*. However, the exact *attL/attR/attS/attB* length required for recombination is still unknown. SEs preferentially incorporate a 6 bp left flank into circular transposition intermediates via a copy-out-like process and insert the 6 bp into the target location *attB*. The 6 bp originating from a donor location is placed at the right flank of a new SE copy as a "footprint" of SE transposition from a specific donor location (15, 16). The movement of 6 bp SE-flanking sequences is consistent with the occurrence of nicking at a specific strand in *attS*, strand exchange with *attB*, and subsequent Holliday junction resolution or replication on the integration intermediate; however, detailed mechanisms of SE integration have not yet been investigated (15, 16). Currently, it remains unclear whether SE is the main genetic mechanism moving *bla*<sub>GMA-1</sub> between genomic locations.

Mechanisms underlying resistome development could provide insights into the emergence of multidrug-resistant bacteria in humans (21). To increase the fundamental knowledge of resistome development in the environment, we addressed the substrate range of GMA-1 and how frequently *bla*$_{GMA-1}$ was linked to SEs in sequences in publicly available genomes.

## RESULTS AND DISCUSSION

### GMA-1 is a group 2c penicillinase

The catalytic motifs in GMA-1 are identical to those found in VHH-1 and CARB-6 (Fig. 1); therefore, GMA-1 was hypothesized to possess characteristics of 2c β-lactamases. To validate this, a DNA segment containing the promoter and coding sequence of $bla_{GMA-1}$ in pAQU1 from *P. damselae* was PCR-amplified, cloned into the low-copy vector pMW218, and introduced into *E. coli* to assess the minimum inhibitory concentration (MIC) of β-lactams. The MICs of penicillin (ampicillin, amoxicillin, piperacillin, and carbenicillin) against *E. coli* were increased after $bla_{GMA-1}$ introduction despite the presence of β-lactamase inhibitors (Table 1). The MIC of cefazolin, a first-generation cephalosporin, was also increased with the introduction of $bla_{GMA-1}$. The MIC against *E. coli* did not increase with the introduction of $bla_{GMA-1}$ for the second, third, and fourth generation cephalosporins, carbapenems, aztreonam, or flomoxef. This indicates that GMA-1, like the CARB family (11), can act on penicillin and partially on cephalosporins, but not the second, third, and fourth generations or other β-lactams. Although clavulanate and sulbactam are reported to have some inhibitory effects on PSE-1 *in vitro* (22), this was not observed at the MIC level for GMA-1. These observations largely support the claim that GMA-1 belongs to functional group 2c.

### Genomes carrying $bla_{GMA-1}$

Previously, we identified two SEs carrying $bla_{GMA-1}$: SE-6945 on pSEA2 and SE-Pda04Ya311 on pAQU1 (15, 16). However, it is unknown whether SE is the main type of mobile DNA element linked to $bla_{GMA-1}$. Since we previously identified pAQU1 by exogenous plasmid capture using mating and sequencing (13), it is unknown whether the original pAQU1 host strain 04Ya311 carried another copy of $bla_{GMA-1}$ and SE in the chromosome, as found in two *V. alfacsensis* strains isolated from the same area in Japan as 04Ya311 (15). To finalize the analysis of 04Ya311, we determined the complete genome sequence of 04Ya311 using ONT long reads and Illumina short reads. Consequently, 04Ya311 was shown to have two chromosomes, pAQU1, and the only copy of SE-Pda04Ya311 on pAQU1.

To identify DNA molecules that acquired $bla_{GMA-1}$ and the associated host taxa, we further searched for nucleotide sequence entries containing $bla_{GMA-1}$ in the NCBI RefSeq and GenBank databases. In total, 39 hits were detected, and subsequent filtering for contigs containing >10 kb on either side of $bla_{GMA-1}$ and deduplication identified 21 filtered RefSeq/GenBank entries, including pAQU1 (Table S1). The host species of $bla_{GMA-1}$ in the final data set are *Vibrio parahaemolyticus*, *V. alfacsensis*, *P. damselae* subsp. *damselae*, *P. damselae* subsp. *piscicida*, *Vibrio alginolyticus*, *Vibrio furnissii*, *Vibrio gangliei*, *Vibrio scophthalmi*, and *Pseudoalteromonas agarivorans*.

### SE-mediated $bla_{GMA-1}$ acquisition by chromosomes and plasmids

To address whether SE is the main type of mobile element disseminating $bla_{GMA-1}$, we searched for the presence of SE core genes (Fig. 2A) in 21 filtered RefSeq/GenBank entries carrying $bla_{GMA-1}$. Aside from the five previously reported RefSeq/GenBank entries containing SE-6945 and SE-Pda04Ya311, six new entries were found to carry SE core genes and $bla_{GMA-1}$: pMT14 from *V. furnissii* (NZ_CP115190.1), pC1579 from *V. alginolyticus* (NZ_MN865127.1), *V. parahaemolyticus* UCM-V493 chromosome 1 (NZ_CP007004.1), *P. agarivorans* Hao 2018 chromosome I (NZ_CP033065.1), *V. scophthalmi* VS-12 chromosome 1 (NZ_CP016307.1), and *V. scophthalmi* FP3289 contig00002 (NZ_MDCJ01000002.1).

Since SEs should copy the 6 bp left flank at the donor location and insert it into the right flank after integration as a 6 bp footprint, independent SE transposition events can be inferred by comparing 6 bp footprint sequences even when the equivalent genomic locations are occupied by the same SE. For example, the 6 bp footprint of SE-Pda04Ya311 on pAQU1 [Fig. 3A(i)] is TATGAG, while it is TACGAA on plasmid pMT14 [Fig. 3A(ii)];

**TABLE 1**  β-Lactam MICs against *E. coli* strains carrying *bla*$_{GMA-1}$

| β-Lactam group | Antimicrobial/inhibitor[a] | *E. coli* HST08 (pMW218) | *E. coli* HST08 (pHY1389) |
|---|---|---|---|
| Penicillin | Carbenicillin | 4 | >512 |
| | Ampicillin | ≤4 | >16 |
| | Ampicillin/sulbactam | ≤4/2 | >16/8 |
| | Amoxicillin/clavulanate | ≤8/4 | >16/8 |
| | Piperacillin | ≤4 | >64 |
| | Piperacillin/tazobactam | ≤4/4 | >64/4 |
| Carbapenem | Imipenem | ≤0.5 | ≤0.5 |
| | Meropenem | ≤0.25 | ≤0.25 |
| Cephalosporin | Cefazolin | ≤1 | 2 |
| | Cefaclor | ≤8 | ≤8 |
| | Cefmetazole | ≤4 | ≤4 |
| | Cefotiam | ≤8 | ≤8 |
| | Cefditoren | ≤1 | ≤1 |
| | Ceftazidime | ≤1 | ≤1 |
| | Cefpodoxime | ≤1 | ≤1 |
| | Ceftriaxone | ≤0.5 | ≤0.5 |
| | Cefotaxime | ≤0.5 | ≤0.5 |
| | Cefoperazone/sulbactam | ≤8/4 | ≤8/4 |
| | Cefepime | ≤1 | ≤1 |
| Monobactam | Aztreonam | ≤1 | ≤1 |
| Oxacephem | Flomoxef | ≤8 | ≤8 |

[a]MICs were determined using the Beckman MicroScan system, except for carbenicillin. Unit of concentration, mg/mL.

therefore, the integration of SE-Pda04Ya311 into *traN* downstream of ancestral forms of pAQU1 and pMT14 likely occurred independently (*bla*$_{GMA-1}$ acquisition patterns C1 and C2 in Table S1). Pattern C2 was also observed in pC1579 from *V. alginolyticus* (Table S1), so pC1579 and pMT14 probably diverged from a common ancestor carrying SE-Pda04Ya311.

An SE-Pda04Ya311 copy was also identified in the chromosomes of *V. scophthalmi* VS-12 and *V. scophthalmi* FP3289 (patterns F1 and F2). This copy integrated in the *rraB* gene [Fig. 3B(i)] has a 6 bp footprint of GATGCC in VS-12 and TATGAG in FP3289 [Fig. 3B(ii)], suggesting that ancestors of these two strains acquired SE-Pda04Ya311 copies from two different donor locations.

SE-VpaV493, a new SE, was identified to be integrated in the *rraB* gene of *V. parahaemolyticus* [Fig. 3C(i), pattern D]. SE-VpaV493 is very similar to SE-Pda04Ya311 at the nucleotide level. They possess identical terminal sequences [Fig. 3A(ii) and C(ii)] and a gene encoding the DUF262-domain-containing protein (RefSeq ID: WP_014386725.1), which is speculated to function as a modification-dependent restriction nuclease (Type IV restriction-modification system) (23). These two SEs differ in the cargo genes encoding hypothetical proteins carried at the right end. SE-PagHao2018C, another new SE, was identified in the PAGA_a3483-quivalent locus of *P. agarivorans* Hao2018 [Fig. 3D(i), pattern E]. It has imperfect inverted repeat motifs (C/C') different from the C/C' of SE-Pda04Ya311 [Fig. 3D(ii)].

We previously reported two *bla*$_{GMA-1}$ acquisition events by the ancestor of pSEA2 and the chromosome of *V. alfacsensis* (patterns A and B in Table S1) (15). Thus, eight independent SE-mediated *bla*$_{GMA-1}$ acquisition events were identified in the currently available RefSeq/GenBank data set. Since nucleotide sequences of SE core genes are considerably different between the three SE lineages (SE-6945, SE-PagHao2018C, SE-Pda04Ya311), we speculate that they independently captured *bla*$_{GMA-1}$ in the past rather than having diversified from one common ancestral SE carrying *bla*$_{GMA-1}$ (Fig. 3E).

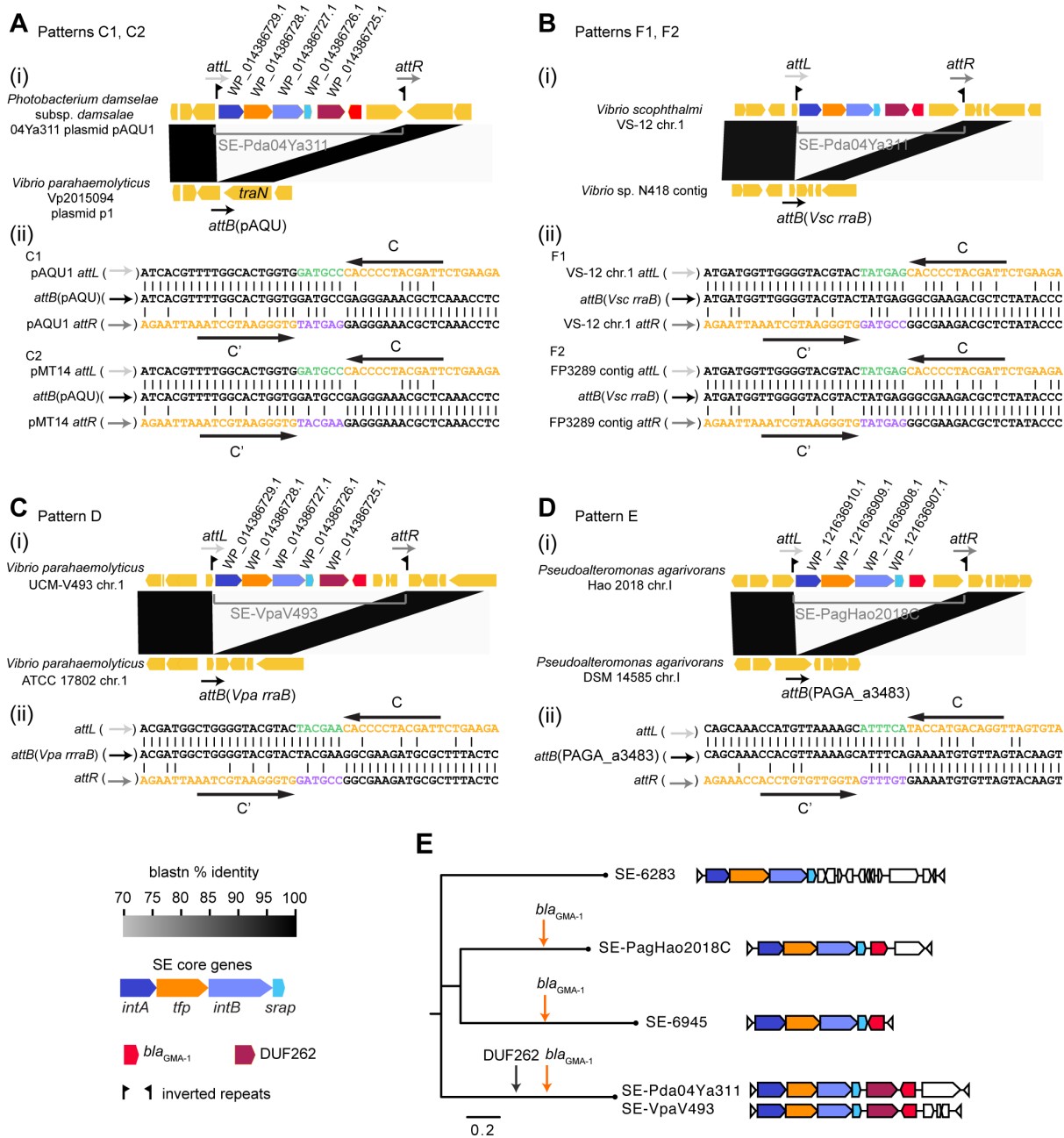

**FIG 3** Delineation of SEs carrying $bla_{GMA-1}$ and their termini. (A) SE-Pda04Ya311 in pAQU1-like plasmids. Top: comparison of genetic structure between two genomes (with and without SE). Bottom: alignment of *attB*, *attL*, and *attR* shown in the upper panel. A putative 6 bp footprint (16) introduced by SE integration is shown in purple. The 6 bp sequence incorporated into the SE circular form is shown in green; SE sequences are shown in orange. Positions of terminal imperfect inverted repeat motifs, C/C′, are indicated by horizontal arrows over the sequences. (B) SE-Pda04Ya311 in *Vibrio* chromosomes. (C) SE-VpaV493 in the *V. parahaemolyticus* chromosome. (D) SE-PagHao2018C in the *Pseudoalteromonas agarivorans* chromosome. (E) Acquisition of $bla_{GMA-1}$ by three SE lineages. The tree was inferred using the maximum likelihood method with iqtree2 according to the aligned concatenated SE core gene products. Diagrams of the genetic organization of SEs are shown next to tip labels. Pentagons are protein-coding sequences. Red arrows on the tree indicate three putative independent $bla_{GMA-1}$ acquisition events by three SE lineages. SE-6283 was previously called Tn*6283* (14).

## SE-independent acquisition of $bla_{GMA-1}$ by chromosomes and plasmids

Out of 21 filtered RefSeq/GenBank entries with $bla_{GMA-1}$, 10 did not contain SE genes. The inspection of $bla_{GMA-1}$ flanks of 10 entries revealed six unique $bla_{GMA-1}$ insertion locations associated with six putatively independent $bla_{GMA-1}$ acquisition events. One

striking example was the translocation of $bla_{GMA-1}$ alone, which was observed in plasmid-like contigs from *V. parahaemolyticus* strains Vb2839 and Vb2840 (GenBank accession numbers: JARJXW010000007.1, JARNUH010000007.1) [Fig. 4(i), pattern K in Table S1]. In these plasmid-like contigs, a 1,483 bp $bla_{GMA-1}$ region was inserted into a hypothetical protein gene (no locus_tag assigned, protein ID ANS55671.1) present in pVPS92-VEB, and it was flanked by 5 bp direct repeats just like the target site duplication (TSD) generated by the transposition of a transposon [Fig. 4(ii)]. The terminal region of the 1,483-bp insert forms 42-bp imperfect inverted repeats (IRU and IRD) [Fig. 4(iii)]. IRU and IRD showed no similarity with terminal inverted repeats of IS elements registered in the ISfinder database (24). Therefore, the acquisition of $bla_{GMA-1}$ alone on the ancestor of these *Vibrio* plasmids probably occurred independently of mobile DNA elements.

The remaining five unique $bla_{GMA-1}$ insertion locations (patterns G, H, I, J, and L in Table S1) are shown in Fig. S1. Four of these were on plasmids (p1, pVgang, plasmid-like contig of PP3, and pP9014), while one was on a chromosome (contig of *V. parahaemo-lyticus* YK32). These $bla_{GMA-1}$ insertion locations are within a variable region where the equivalent region in the closest plasmid or chromosome also contains a different insertion type; therefore, the precise borders of mobilized units of $bla_{GMA-1}$ could not be determined by genomic comparisons.

For example, $bla_{GMA-1}$ on plasmid p1 [Fig. S1A(i)] is flanked by a tyrosine recombinase gene that was identified to not be an integron integrase gene according to Integron-Finder 2 (25). Blastn searches in ISfinder (24) revealed that two IS*Shf9*-like ISs were located in the upstream region of $bla_{GMA-1}$ on plasmid p1. However, IS*Shf9*-related terminal inverted repeats, termed IR1–IR5, did not bracket $bla_{GMA-1}$. A comparison of p1 with pVPSD2016-2 revealed that pVPSD2016-2 has a 49-bp segment instead of the 21,695-bp $bla_{GMA-1}$-containing segment in p1. pVgang from *V. gangliei* also has $bla_{GMA-1}$ and IS*Shf9*-like elements, but it lacks the integrase gene located next to $bla_{GMA-1}$ on p1 (Fig. S1B). The mechanisms by which p1 and pVgang ancestors acquired $bla_{GMA-1}$ (patterns J and H) could not be resolved.

The plasmid-like contig of *P. damselae* subsp. *piscicida* PP3 was similar to plas1 in genetic organization (Fig. S1C), and plas1 has an IS*Alg*-like IS and one coding sequence in the $bla_{GMA-1}$ location-equivalent region. The occurrence of $bla_{GMA-1}$ alone translocation was speculated for the PP3 contig, but inverted repeat motifs (IRU/IRD) and associated TSD found in pattern K [Fig. 4(iii)] were not identified in the PP3 contig.

$bla_{GMA-1}$ on pP9014 from *P. damselae* subsp. *piscicida* (pattern I) and that on the YK32 chromosome (pattern L) were embedded in variable regions (Fig. S1D and E). No IS or site-specific recombinase gene was identified in these regions. No DNA-nicking enzymes appeared to be repeatedly associated with $bla_{GMA-1}$ acquisition by six distinguishable DNA molecules without SEs (patterns G, H, I, J, K, and L).

## $bla_{GMA-1}$ context

Comparative genomic analysis revealed nine unique $bla_{GMA-1}$ insertion locations (Fig. 3 and 4; Fig. S1). To obtain insights into the rules implicated in $bla_{GMA-1}$ translocation, $bla_{GMA-1}$-flanking sequences were inspected (Fig. 5). The sequences are shared among all nine $bla_{GMA-1}$ locations up to 251 bp upstream and 109 bp downstream of $bla_{GMA-1}$, after which they diversify. Inverted repeats and putative promoter motifs, identified by SAPPHIRE (26) (*P*-value, 0.0033), are present in the upstream region of $bla_{GMA-1}$ (Fig. 5A). One inverted repeat (green in Fig. 5A) was identical in all sequences, whereas the other had differences in sequence. Inverted repeats or transcriptional terminator motifs were not detected downstream of $bla_{GMA-1}$ (Fig. 5B). Therefore, a 1,173 bp conserved segment, starting from the center of inverted repeats in $bla_{GMA-1}$ upstream, might have repeatedly moved into multiple locations by an unknown molecular mechanism. This $bla_{GMA-1}$-containing segment may have remained unchanged due to selection to retain a function and expression level harmless for hosts.

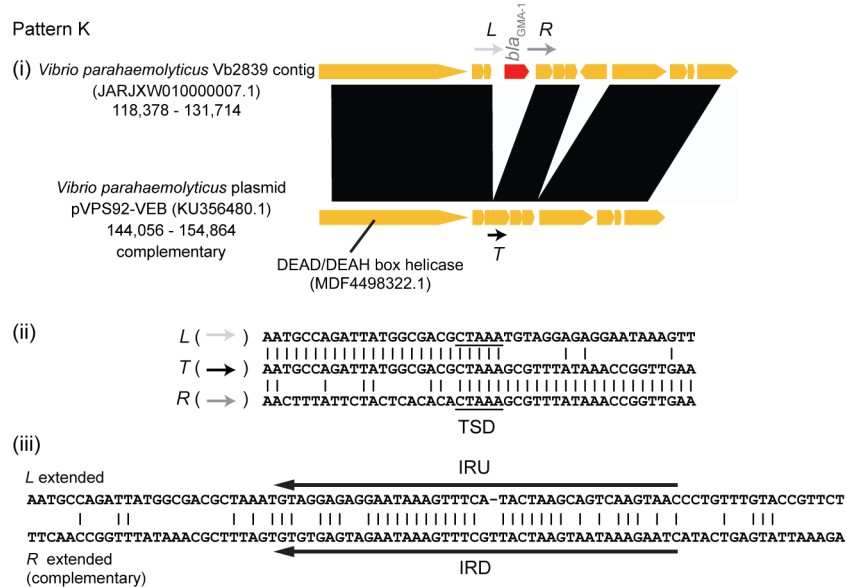

**FIG 4** Translocation of *bla*GMA-1 alone. Top: comparison between two genomes (one with *bla*GMA-1 insertion alone and the other without). *L*, left border; *R*, right border; *T*, target sequence. Middle: alignment of *L*, *R*, and *T*. Bottom: imperfect inverted repeats, IRU and IRD, found in the terminal region of the *bla*GMA-1 insert. The TSD is underlined.

## Conclusions

This study demonstrates that *bla*GMA-1 encodes a group 2c narrow-spectrum penicillinase. Twelve unique *bla*GMA-1 insertion locations were found in RefSeq/GenBank entries of complete sequences or linear contigs >20 kb; seven are in plasmids, and five are in the chromosomes. *bla*GMA-1-flanking sequence analysis revealed a total of eight distinct SE-mediated *bla*GMA-1 acquisition events (five by chromosomes, three by plasmids) and six SE-independent *bla*GMA-1 acquisition events (one by chromosome, five by plasmids) without involving specific mobile elements, thus reinforcing the concept that the SE, putatively hitchhiking on plasmids or integrative and conjugative elements, is the main type of mobile DNA element disseminating *bla*GMA-1.

## MATERIALS AND METHODS

### Strains and media

*P. damselae* subsp. *damselae* strain 04Ya311 harboring pAQU1 (13) was cultured at 30°C in a BBL Brain Heart Infusion (BD, Flanklin Lake, NJ, USA) supplemented with up to 2% NaCl. *E. coli* HST08 [F⁻, *endA1*, *supE44*, *thi-1*, *recA1*, *relA1*, *gyrA96*, *phoA*, Φ80d*lacZ*ΔM15, Δ(*lacZYA-argF*)U169, Δ(*mrr-hsdRMS-mcrBC*), Δ*mcrA*, λ⁻] (Takara Bio Inc., Shiga, Japan) was cultured in a Difco LB broth, Lennox (BD), or LB agar plate at 37°C for cloning steps and in Difco Muller–Hington broth (BD) at 35°C for susceptibility testing.

### Genome sequencing of 04Ya311

The genomic DNA of 04Ya311 was purified from 4 mL of an overnight culture with a Qiagen DNeasy Blood & Tissue Kit (Qiagen, Hilden, Germany). A whole-genome sequencing of 04Ya311 was performed on a MiniSeq (Illumina, Inc., San Diego, CA, USA) platform with a MiniSeq High Output Reagent Kit (300 cycles) and a MinION (Oxford Nanopore Technologies Ltd., Oxford, UK) platform with an R9.4.1 flow cell. A library for Illumina sequencing (paired-end, insert size of 500–900 bp) was prepared with a Nextera XT DNA Library Prep Kit, and a library for MinION sequencing was prepared with a Rapid

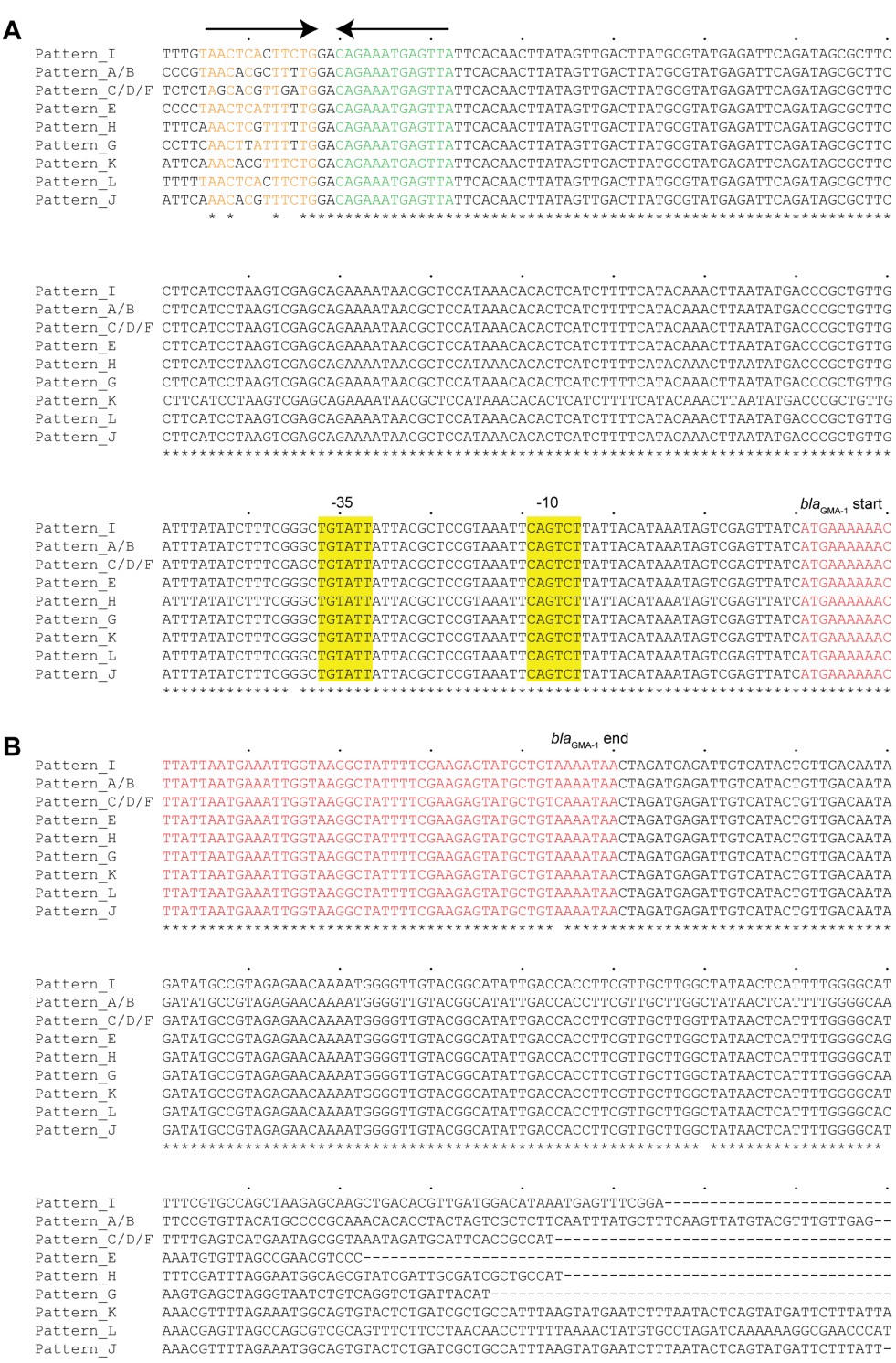

**FIG 5** Nucleotide sequences of *bla*<sub>GMA-1</sub>-flanking regions at nine *bla*<sub>GMA-1</sub> insertion locations. The locations are associated with *bla*<sub>GMA-1</sub> acquisition patterns. Pattern_A/B indicates a location within SE-6945, whereas Pattern_C/D/F indicates a location within SE-Pda04Ya311 and SE-VpaV493. Fully conserved nucleotides are indicated by asterisks below the sequences. *bla*<sub>GMA-1</sub>-coding sequences are indicated by a red font. (A) *bla*<sub>GMA-1</sub> upstream regions. Inverted repeats are highlighted. Promoter motifs (−35 box, −10 box) were predicted by SAPPHIRE (26). (B) *bla*<sub>GMA-1</sub> downstream regions. The sequences originated from the following RefSeq/GenBank accession numbers: pattern_I, NC_012919.1; pattern_A/B, NZ_AP019849.1; pattern_C/D/F, NZ_CP007004.1; pattern_E, NZ_CP033065.1; pattern_H, NZ_AP021871.1; pattern_G, NZ_SRHT02000006.1; Pattern_K, JARJXW010000007.1; pattern_L, NZ_CP115190.1; and pattern_J, NZ_CP080480.1.

Barcoding Kit (SQK-RBK004). ONT reads were base-called with Guppy v5.0.11 in super-accuracy mode. Long read-only genomic assembly was performed with Trycycler v. 0.5.3 (27), following the flow recommended by a software developer (https://github.com/rrwick/Trycycler/wiki); the assembly was polished using Illumina reads and Pilon v. 1.23 (28).

## Cloning of $bla_{GMA-1}$ and antimicrobial susceptibility testing

$bla_{GMA-1}$ and the promoter region on pAQU1 were PCR-amplified with primers mcs_blaGMA_F: 5′- TCGAGCTCGGTACCCCACACTCATCTTTTCATACAAAC-3′ and mcs_blaGMA_R: 5′-CTCTAGAGGATCCCCTTATTTGACAGCATACTCTTC-3′ and KOD One PCR Master Mix (Toyobo, Osaka, Japan). The low-copy number cloning vector pMW218 ($oriV_{pSC101}$, Km$^r$; Nippon Gene Co., Ltd., Toyama, Japan) was PCR-amplified with primers pMW_mcs_F: 5′-GGGTACCGAGCTCGAATTCGTA-3′ and pMW_mcs_R: 5′-GGGGATCCTCTAGAGTCGACCT-3′. The two PCR products were column-purified and combined using a NEBuilder HiFi DNA assembly cloning kit (New England Biolabs, Ipswich, MA, USA). The reaction mixture was introduced into competent *E. coli* HST08 cells, and transformants were selected on LB agar with 20 µg/mL kanamycin. Plasmids were purified from transformants, and the insert was determined by Sanger sequencing. The recombinant plasmid carrying $bla_{GMA-1}$ with the authentic promoter was named pHY1389. For antimicrobial susceptibility testing, *E. coli* HST08 clones carrying either pMW218 or pHY1389 were first streaked on an LB agar plate and incubated at 37°C overnight. Subsequently, a single colony was picked, diluted with a MicroScan prompt inoculation system-D, and inoculated into the MicroScan Neg MIC EN 2J panel in a Renok rehydrating inoculator (Beckman Coulter, Inc., Brea, CA, USA). The MIC panel was incubated at 35°C for 18 h in the MicroScan WalkAway40 plus system (Beckman Coulter, Inc.). As carbenicillin is not included in the MicroScan panel, its MIC was determined by a standard microdilution method following Clinical & Laboratory Standards Institute M07. Carbenicillin was tested at 4, 8, 16, 32, 64, 128, 256, and 512 µg/mL. Tests were performed three times on three different days. All three independent experiments gave the same MIC results.

## Phylogenetic analysis

Protein sequences of 12 characterized class A β-lactamases were retrieved from the NCBI RefSeq database. Sequences were aligned using the structure-aware alignment program PROMAL3D at the default parameter setting (17). A phylogenetic tree was constructed using IQ-TREE v2.1.3 with the maximum likelihood method and "-T AUTO--model-joint NONREV" options (29).

## Genome comparisons and identification of mobile genetic elements

The RefSeq/GenBank ID linked to the NCBI protein ID (WP_012774820.1) of GMA-1 was searched for at the NCBI identical protein groups database (https://www.ncbi.nlm.nih.gov/ipg). In total, 39 RefSeq and GenBank hits (19 September 2023) were retrieved. These sequences from the same strain were deduplicated, and contigs carrying >10 kb segments from each end of $bla_{GMA-1}$ were selected. The final data set contained 21 sequences of 7 finished chromosomes, or chromosome-like contigs, and 14 finished plasmids, or plasmid-like contigs. To compare the 21 filtered sequences with other sequences of closely related replicons without $bla_{GMA-1}$, blastn was performed on the NCBI blast website (https://blast.ncbi.nlm.nih.gov/Blast.cgi) using the sequences of $bla_{GMA-1}$-flanking regions or plasmid genes as queries. Subsequently, sequences of RefSeq/GenBank hits without $bla_{GMA-1}$, related to $bla_{GMA-1}$-carrying replicons, were retrieved from NCBI for comparative genomics analysis.

RefSeq/GenBank files encoding GMA-2, another GMA variant that differed from GMA-1 by six amino acids, were also searched for in the NCBI database. The only GMA-2 hit was the chromosomal contig (RefSeq ID: NZ_VXDD01000003.1) of *Vibrio fortis* S7-72,

which was not associated with SE. This was not further analyzed due to the lack of information for $bla_{GMA-2}$ movement analysis.

An SE unit was defined by the presence of (i) a cluster of four SE core genes encoding the following products in a specific order: IntA, tyrosine recombinase carrying the catalytic RHRY motif; Tfp, tyrosine recombinase fold protein, which exhibits similarity with tyrosine recombinase at the protein structure level but lacks the RHRY motif; IntB, long tyrosine recombinase carrying the catalytic RHRY motif; and Srap, SE-associated recombination auxiliary protein, with (ii) approximately 19 bp of imperfect inverted repeats (16). InterProScan (https://www.ebi.ac.uk/interpro/about/interproscan) was used to confirm the presence of tyrosine recombinase-related domains (InterPro IDs: IPR013762, IPR011010) in IntB and Tfp candidates. The presence of the catalytic RHRY motif of tyrosine recombinases (30) in IntA and IntB homologs was confirmed by manual inspection of product sequences. Jpred4 (31) was used to identify a Srap homolog similar to the Srap of SE-6945 and SE-6283 at the secondary structural level. Termini of SE were identified by genome comparisons between SE-positive and SE-free genomes; SE termini were identified when: (i) split *attB* sequences were found in either *attL* or *attR,* and (ii) putative terminal sequences of SE could form imperfect inverted repeats after the removal of 6 bp from the 3′ end of the putative SE insertion region.

Coding regions of integron integrase and gene cassettes were searched using IntegronFinder 2.0 (25). Known insertion sequences were searched using ISfinder (24). Pairwise genome sequence comparisons were performed with the BLASTn or BLASTp function (32) implemented in GenomeMatcher (33). Promoter and transcriptional terminator predictions were performed using SAPPHIRE (26) and ARNold (34), respectively.

## ACKNOWLEDGMENTS

We thank Wataru Hayashi at the National Institute for Infectious Diseases (NIID) for the helpful discussion and Shuichi Mori at the NIID for the use of laboratory equipment. We thank Satoyo Wakai at the NIID for the operation of the Microscan WalkAway40 system. Computation was supported by the supercomputer system SHIROKANE at the Human Genome Center at the Institute of Medical Sciences at the University of Tokyo (IMSUT). We thank Enago for language editing.

This work is supported by JSPS KAKENHI under grant numbers 18K05790 (L.N., H.Y.), 22K05790 (L.N., H.Y.), and 23K06556 (M.S.); AMED under grant numbers JP23gm1610003, JP23fk0108642, JP23fk0108665, JP23fk0108683, JP23wm0325037, JP23wm0225022, and JP23wm0225029 (M.S.); Mishima Kaium Memorial Foundation (H.Y.); and Ohsumi Frontier Science Foundation (M.S.).

## AUTHOR AFFILIATIONS

[1]Antimicrobial Resistance Research Center, National Institute of Infectious Diseases, Higashimurayama, Tokyo, Japan
[2]Faculty of Human Life Sciences, Shokei University, Kumamoto, Japan

## AUTHOR ORCIDs

Hirokazu Yano http://orcid.org/0000-0001-5144-3459
Masato Suzuki http://orcid.org/0000-0001-8975-2193
Lisa Nonaka http://orcid.org/0000-0003-0881-4626

## FUNDING

| Funder | Grant(s) | Author(s) |
| --- | --- | --- |
| MEXT | Japan Society for the Promotion of Science (JSPS) | 18K05790, 22K05790 | Lisa Nonaka |
| Ohsumi Frontier Science Foundation (OFSF) | | Masato Suzuki |

| Funder | Grant(s) | Author(s) |
|---|---|---|
| Mishima Kaiun Memorial Foundation | | Hirokazu Yano |
| Japan Agency for Medical Research and Development (AMED) | JP23gm1610003, JP23fk0108642, JP23fk0108665, JP23fk0108683, JP23wm0325037, JP23wm0225022, JP23wm0225029 | Masato Suzuki |
| MEXT \| Japan Society for the Promotion of Science (JSPS) | 23K06556 | Masato Suzuki |
| MEXT \| Japan Society for the Promotion of Science (JSPS) | 18K05790, 22K05790 | Hirokazu Yano |

## AUTHOR CONTRIBUTIONS

Hirokazu Yano, Conceptualization, Formal analysis, Funding acquisition, Investigation, Visualization, Writing – original draft, Writing – review and editing | Masato Suzuki, Funding acquisition, Methodology, Writing – original draft, Writing – review and editing | Lisa Nonaka, Conceptualization, Funding acquisition, Investigation, Resources, Writing – review and editing

## DATA AVAILABILITY

The complete genome sequence of 04Ya311 was deposited under DDBJ/GenBank/EMBL accession numbers AP026780.1, AP026781.1, and AP026782.1 for chromosome 1, chromosome 2, and plasmid pAQU1, respectively. The raw sequence reads generated in this study were deposited under the SRA/DDBJ sequence read archive accession number DRA014815.

## ADDITIONAL FILES

The following material is available online.

### Supplemental Material

**Fig. S1 (Spectrum02589-23-s0001.docx).** $bla_{GMA-1}$ insertion locations without SE genes.
**Table S1 (Spectrum02589-23-s0002.xlsx).** RefSeq and GenBank entries containing $bla_{GMA-1}$.

### Open Peer Review

**PEER REVIEW HISTORY**
**(review-history.pdf).** An accounting of the reviewer comments and feedback.

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
