## [Reviewer comments · Microbiology Spectrum]

Microbiology Spectrum

Mobile class A β -lactamase gene *bla*_{GMA-1}

Hirokazu Yano, Masato Suzuki, and Lisa Nonaka

Corresponding Author(s): Hirokazu Yano, National Institute of Infectious Diseases

Review Timeline:

Submission Date:	June 22, 2023
Editorial Decision:	September 15, 2023
Revision Received:	November 7, 2023
Accepted:	November 12, 2023

Editor: Daria Van Tyne

Reviewer(s): The reviewers have opted to remain anonymous.

Transaction Report:

DOI: <https://doi.org/10.1128/spectrum.02589-23>

September 15, 2023

Dr. Hirokazu Yano
National Institute of Infectious Diseases
Antimicrobial Resistance Research Center
4-2-1 Aobacho
Higashimurayama, Tokyo 189-0002
Japan

Re: Spectrum02589-23 (Mobile class A β -lactamase gene *bla*_{GMA-1})

Dear Dr. Hirokazu Yano:

Thank you for submitting your manuscript to Microbiology Spectrum. Your study was reviewed by two experts, and I would now like you to revise your manuscript in line with their feedback.

Link Not Available

Sincerely,

Daria Van Tyne

Journals Department
Reviewer comments:

Reviewer #1 (Comments for the Author):

This manuscript has a few different aspects, describing experiments to characterize a recently identified beta-lactamase named GMA-1 first found on a strand-biased circularizing integrative element (SE) and analysis of available contexts of this gene, identifying additional SE that carry it and also contexts outside SE. The data are of interest, the experiments to characterize GMA-1 appear sound and the conclusions are reasonable. However, I found both the text and figures relating to the results of sequence analyses difficult to follow. From a quick look at some of the sequences, I think these parts are unnecessarily complicated and could be simplified - I have given some suggestions on this in my comments below, including for some reorganization - describing characterization of GMA at the start works, but then describing the new SE, then the "patterns" might

work better. Only sequences up to September 2022, which is now almost a year ago, are included in the analysis and checking for updates might add to the study. NCBI also now lists a blaGMA-2 variant, which could also be looked at or at least commented on. Some minor rewording would also improve clarity, with suggestions given below.

1) Introduction

As SE elements have been identified only very recently, clearly describing their characteristics would be helpful to readers e.g., move some information from Fig 2 legend to the main text and/or refer to this figure on the Introduction? Also see comments of Fig. 2 below.

Lines 64, 237 - att sites seem to be mentioned only on Line 237 and in Figure 2 legend - it is not clear how these relate to the inverted repeats.

Line 36 - "transporters" "porins" "of beta-lactamase genes"

Line 38 - suggest "beta-lactamases of clinical concern include.." ESBLs are not carbapenemases, so this needs clarifying.

Line 45 - "in class 1 integrons"

Line 49 "and carbapenems"

Line 50 - suggest "several *Vibrio* spp."

Line 52 - should this be aquaculture?

Line 53 - suggest "reported three conjugative... plasmids from". second "three" is not needed.

Line 57-8 - "Among class A beta-lactamases" not really needed. Suggest "both over the whole sequence and at motif level".

Line 61 - "E. coli transconjugants containing..."

Line 62 - "plasmid" is not needed before pAQU1 - the "p" indicates that it is a plasmid (also lines 92, 95, 100, 109, 112, 142, 174, 243 etc) - and the species could be indicated in parentheses.

Line 64 - "their termini".

Lines 65-7 - this could be explained more clearly.

Line 70 - "resistome development"?

2) Characterization of GMA-1 - Lines 76-90,

I have only a few suggestions for rewording here.

Lines 82-4 - strictly speaking, it is the antibiotic that has the MIC against the isolate, so this should be reworded.

Line 78 - "of 2c beta-lactamases" - no "the" needed - also start of line 85.

Line 79 - "from *P. damsela*"?

Line 80 - "low-copy vector"?

Line 87 - "cephalosporins, but not 2nd, 3rd or 4th generation"

Fig. 1 legend - the tree is for the whole proteins or the motifs? It's not clear what is meant by "shown by their terminal nodes". "Functional groups"?

3) Identification of SE

I would suggest combining information on Lines 116-129 and 133-40 and placing after Lines 91-106, first describing the SE identified here, before going on to talk about the patterns.

Lines 95-8 - I would suggest making it clear here that Refs 13 and 15 are your previous work and this part was essentially to finalize analysis of 04Ya311.

Lines 102-6 - if there are papers associated with the accession numbers they should be cited, at least in Table S1.

Lines 121-7 - the nucleotide sequences of various SE appear highly related, so I would suggest referring to these rather than "gene product sequences" here - if the nucleotide sequences are highly related it follows that the protein products will be too.

Line 123 - what are these cargo genes and where are they in relation to blaGMA-1? This might be most easily shown on a figure (see comments on Figure 2 below).

pSEA-1 Table S1 lists NZ_LC081338.1 as a duplicate of NZ_AP024167.1, with both plasmids named as pSEA1, but the sequences are slightly different lengths. Are both correct? If not, maybe the incorrect one could be removed from GenBank to reduce confusion.

Line 125 - are the SE names correct here? Table S1 and elsewhere in the text have VpaV493, not VdaVS492, and SE-VcVS12C, not SE-VcVS12B.

Fig. 2 - I would suggest changing this to better show the relationships between all SE analysed here and incorporate the information on Lines 133-40. Diagrams are not really needed for the sequences without SE - the accessions could just be stated. Diagrams for the previously-characterized SE also analysed here could be included, with all diagrams aligned and SE names added. Variants of the intA, tnp intB and srp genes could each be shown e.g. in different shades of the same colour, as the different WP_ accessions are not easy to distinguish at a glance. The att site alignments for the new SE could be shown separately.

Fig. 3A - I find this format very difficult to interpret and I don't think it is useful to include the non-SE blaGMA-1 contexts here - this figure is not really needed if the suggested changes to Fig. 2 are made and Fig. 3B could be shown separately or maybe with Fig. 2 or Fig. 4? Legend, Line 389 - "similarity" and "identity" have specific meanings in relation to proteins

Line 91 - suggest "SE are the main type of mobile element..."

Lines 93, 172 - not clear what is meant by "primal" here?

Line 98 - suggest "to investigate this"

Line 100 - this is unclear - the only copy is on pAQU1?

Line 103 - suggest ">5 kb on either side of blaGMA-1" if this is what is meant?

Lines 103-4 - "13 entries including pAQU1" might work better.
Lines 104-6 - if all species are listed then "included" should be removed.

4) blaGMA-1 contexts

I would suggest combining information on Lines 107-116, 130-33 and 141-165 to describe the different "patterns". This information is currently presented in a way that I think is overly complicated and confusing.
From Table S1, "patterns" 1-6 just seem to correspond to different SE and this could be made much clearer in the text once all of the SE have been described
Lines 109-111 - these are all identical SE-6945, so it is not clear why they are considered two "patterns"?
Lines 109-112 - are the whole plasmids and chromosome closely related or just the blaGMA-1 contexts?
Lines 112-6 - similarly these are all SE-Pda04Ya311 and this could just be stated.
Lines 118-29 - if this information is moved, as suggested above, patterns 4-6 could just be equated with the relevant SE. Also consider if the "patterns" are really different between highly related SE with some different segments?
Lines 130-33 - see comments on Fig. 3 above.
Lines 141-55 - I think that this analysis has some problems. It is not clear why integron components might be expected and the fact that IS and Tn were searched for should also be mentioned before describing the elements that were found. I am not sure that the presence of any of these transposases is actually relevant - the whole IS elements need to be identified and checked for TSD etc, to see if they are just simple insertions. A quick look suggests that some are not close to blaGMA-1 and maybe associated with different resistance genes.
Are the IS actually ISShrf9, ISVsa5 and ISKpn13 or relatives (cut off is 95% identical nt sequences in ISFinder). More information about the types of plasmids and any relationships between them should also be included.
Lines 159-63 - it might be good to make it clearer here that one IR is identical in all sequences but the other has differences.
Fig. 4 - it is not immediately clear what these six sequences represent, without referring back to other figures, or why only these sequences are included - using only the accessions to identify them is not informative.
Could the extra T in the motif in CP007004 be an error? If the reads are available, it would be possible to check this by read mapping.
The legend should explain what * indicate. I would suggest using dots to mark every 10th base rather than numbering all positions.
How were the promoter motifs identified? I couldn't find thsi in methods. The -10 motif seems extended?
Line 132 - suggest "six of these nine locations are associated with SE"
Line 133 - is there sufficient evidence to generalise beyond the current dataset?
Line 146 - not clear what "respectively" refers to here?
Line 165 - how would this suppress cost?

5) Methods

Line 178 - this is one strain? If so "...Japan) was cultured"
Lines 193-7 - primer sequences are usually given in uppercase.
Lines 200-2 - how was this determined, i.e. was the recombinant plasmid sequenced? If so, this should be stated.
Line 208 - "is not included" "by standard microdilution"
Line 210 - "on three"
Line 222 - delete "of".
Line 225, 232 - the listed names are of genes, not proteins.
Line 235 - "the termini of SE were"?

6) Suggested minor wording changes to improve clarity etc in other parts of the manuscript

Lines 12-3 - genomic data presumably revealed the gene, not the protein?
Lines 14-6 might be better before Lines 12-3.
Lines 17,30 - suggest "in sequences in..."
Line 20 - "...2c, narrow spectrum beta-lactamases." Also see comments above on number of "patterns"
Line 27 - suggest delete "on class A beta-lactamases"
Line 167 - "demonstrates", again see comments above on number of patterns,
Lines 169-70 - suggest "the chromosome"
Line 171 - "hitchhiking on"
References - Line 279 - the whole aac gene names should be in italics. "beta" in several reference titles probably should be changed to the beta symbol.
Line 364 - "except for carbenicillin"

7) Table S1

Column B - heading should be "strain/plasmid"?
Column C - what does "not open to public" mean - not listed in GenBank? Can this information be obtained from an associated paper or is it suggested by the submitter's location?
Column G "franks" should be "flanks".

Reviewer #2 (Comments for the Author):

The manuscript "Mobile class A β -lactamase gene blaGMA-1" by Yano et al. describes the gammaproteobacterial mobile class A B-lactamase (GMA) gene and how it relates to resistance. The authors also characterize this target through a few approaches. The manuscript has good grammatical structure and details an interesting area of antimicrobial resistance research.

Opportunities:

This manuscript is framed around the genomic/gene components of antimicrobial resistance, however the research detailed in the methods section does not reflect this. I found myself looking for specific information on multiple genomes/isolates and could only identify 1 that was sequenced.

In addition, many of the questions answered in the manuscript are reported as defining features that are already known in the introduction. While this confirms the results, there should be a greater scope to support the research.

There are some interesting areas to explore and the overall manuscript could be strengthened with a broader scope.

Staff Comments:

Preparing Revision Guidelines

Please return the manuscript within 60 days; if you cannot complete the modification within this time period, please contact me. If you do not wish to modify the manuscript and prefer to submit it to another journal, please notify me of your decision immediately so that the manuscript may be formally withdrawn from consideration by Microbiology Spectrum.

Re: Spectrum02589-23 (Mobile class A β -lactamase gene *bla*_{GMA-1})

Dear Dr. Hirokazu Yano:

Thank you for submitting your manuscript to Microbiology Spectrum. Your study was reviewed by two experts, and I would now like you to revise your manuscript in line with their feedback.

<https://spectrum.msubmit.net/cgi-bin/main.plex?el=A3QF1CKqX1A4CRkJ2F6A9ftdWeH9VLii8ezn1yIntZmK4wZ>

Sincerely,

Daria Van Tyne

Journals Department
Reviewer comments:

Reviewer #1 (Comments for the Author):

This manuscript has a few different aspects, describing experiments to characterize a recently identified beta-lactamase named GMA-1 first found on a strand-biased circularizing integrative element (SE) and analysis of available contexts of this gene, identifying additional SE that carry it and also contexts outside SE. The data are of interest, the experiments to characterize GMA-1 appear sound and the conclusions are reasonable. However, I found both the text and figures relating to the results of sequence analyses difficult to follow. From a quick look at some of the sequences, I think these parts are unnecessarily complicated and could be simplified - I have given some suggestions on this in my comments below, including for some reorganization - describing characterization of GMA at the start works, but then describing the new SE, then the "patterns" might work better. Only sequences up to September 2022, which is now almost a year ago, are included in the analysis and checking for updates might add to the study. NCBI also now lists a blaGMA-2 variant, which could also be looked at or at least commented on. Some minor rewording would also improve clarity, with suggestions given below. (Response) We thank reviewer 1 for their interest in this work and suggestions. We increased the dataset and split the comparative genomics parts into two sections: (i) SE-mediated blaGMA-1 acquisition and (ii) SE-independent blaGMA-1 acquisition. The second part was also explained further. A new putative blaGMA-1 translocation event was discovered, which was accompanied by the 5-bp TSD generation. The message that "the SE are the main type of mobile element transmitting *bla*_{GMA-1}" remains the same.

We searched for a *bla*_{GMA-2} variant in the RefSeq/GenBank database; however, there was only one DNA molecule carrying *bla*_{GMA-2}, and *bla*_{GMA-2} was not associated with SE. In this study, we stated that "RefSeq/GenBank files encoding GMA-2, another GMA variant that differed from GMA-1 by six amino-acids, were also searched for in the NCBI database. The only GMA-2 hit was the chromosomal contig (RefSeq ID: NZ_VXDD01000003.1) of *Vibrio fortis* S7-72, which was not associated with SE. This was not further analyzed due to the lack of information for *bla*_{GMA-2} movement analysis." (line 289 in the revised manuscript)

1) Introduction

As SE elements have been identified only very recently, clearly describing their characteristics would be helpful to readers e.g., move some information from Fig 2 legend to the main text and/or refer to this figure on the Introduction? Also see comments of Fig. 2 below.

(Response) We thank reviewer 1 for their suggestions. In the revised manuscript, we explained the genetic organization and 6-bp flanking sequences of SE, and mentioned terms, such as *attL*, using Fig. 2 in the Introduction section. The previous Fig. 2 was updated as the new Fig. 3.

Lines 64, 237 - att sites seem to be mentioned only on Line 237 and in Figure 2 legend - it is not clear how these relate to the inverted repeats.

(Response) The following sentence was added to the Introduction section: "The border regions between SE and the host genome, including inverted repeats, are called *attL* and *attR*. The C and C' joint on the circular SE copy is termed *attS*, whereas the target location in the host genome is termed *attB*. The exact *attL/attR/attS/attB* length required for recombination is still unknown." (line 72-75)

Line 36 - "transporters" "porins" "of beta-lactamase genes"

(Response) We modified the words as suggested. (line 41)

Line 38 - suggest "beta-lactamases of clinical concern include.." ESBLs are not carbapenemases, so this needs clarifying.

(Response) We thank reviewer 1 for bringing this issue to our notice. We have rephrased the sentence as follows: "Several clinically significant β -lactamases fall into class A group 2be (commonly known as extended-spectrum β -lactamases: ESBLs), class A group 2f (carbapenemases), and class B group 3a or 3b metallo- β -lactamases (MBLs) capable of degrading most of the currently available synthetic β -lactams."

We identified additional incorrect group assignments in Fig. 1; carbapenemases should be classified under group "2f." These group assignments were verified based on Beta-lactamase database (BLDB).

Line 45 - "in class 1 integrons"

(Response) We added "in class 1." (line 48)

Line 49 "and carbapenems"

(Response) We modified words as suggested. (line 54)

Line 50 - suggest "several *Vibrio* spp."

(Response) We modified words as suggested. (line 55)

Line 52 - should this be aquaculture?

(Response) We changed "for agricultural purpose" to "for aquaculture." (line 57)

Line 53 - suggest "reported three conjugative... plasmids from". second "three" is not needed.

(Response) We rephrased the sentence as suggested. (line 58)

Line 57-8 - "Among class A beta-lactamases" not really needed. Suggest "both over the whole sequence and at motif level".

(response) We rephrased the sentence as suggested (line 62). "Among class A beta-lactamases" has been deleted.

Line 61 - "E. coli transconjugants containing..."

(response) We corrected the sentence. (line 66)

Line 62 - "plasmid" is not needed before pAQU1 - the "p" indicates that it is a plasmid (also lines 92, 95, 100, 109, 112, 142, 174, 243 etc) - and the species could be indicated in parentheses.

(Response) We omitted the term "plasmid" before plasmid names throughout the manuscript.

Line 64 - "their termini".

(Response) We changed "the termini" to "their termini." (line 72)

Lines 65-7 - this could be explained more clearly.

(Response) In this study, we found a variation in the sequences of 6-bp right flanks of SEs integrated into the same target location on pAQU1-type plasmids (indicating two independent SE integration events in very closely related plasmids). In the revised manuscript, we explained the 6-bp right flank as follows: "SEs preferentially incorporate a 6-bp left flank into circular transposition intermediates via a copy-out like process and insert the 6 bp into the target location *attB*. The 6 bp originating from a donor location is placed at the right flank of a new SE copy as a "footprint" of SE transposition from a specific donor location (ref 15, 16)".

Line 70 - "resistome development"?

(Response) We changed "developments" to "development." (line 85)

2) Characterization of GMA-1 - Lines 76-90,

I have only a few suggestions for rewording here.

Lines 82-4 - strictly speaking, it is the antibiotic that has the MIC against the isolate, so this should be reworded.

(Response) We rephrased the sentence as follows: "The MICs of penicillin (amoxicillin, ampicillin, piperacillin, and carbenicillin) against *E. coli* are increased after *bla*_{GMA-1} introduction despite the presence of β -lactamase inhibitors (Table 1). The MIC of cefazolin, a 1st generation cephalosporin, was also increased with *bla*_{GMA-1}. The MIC against *E. coli* did not increase with *bla*_{GMA-1} for the 2nd, 3rd, and 4th generation cephalosporins, carbapenems, aztreonam, or flomoxef." (line 97)

Line 78 - "of 2c beta-lactamases" - no "the" needed - also start of line 85.

(Response) "The" was omitted from the two sentences as suggested.

Line 79 - "from *P. damsela*"?

(Response) We used "in pAQU1 from *P. damsela*" as suggested. (line 95)

Line 80 - "low-copy vector"?

(Response) We used "low-copy vector" as suggested. (line 96)

Line 87 - "cephalosporins, but not 2nd, 3rd or 4th generation"

(Response) We used "cephalosporins, but not the 2nd, 3rd, 4th generation, or other β -lactams" as language editor suggested so. (line 103)

Fig. 1 legend - the tree is for the whole proteins or the motifs? It's not clear what is meant by "shown by their terminal nodes". "Functional groups"?

(Response) We used "functional groups" as suggested. We rephrased the sentence as follows: "The tree is inferred based on the PROMAL3D (27) alignment of whole amino acid sequences. Three conserved motifs of serine β -lactamases (35) were identified by inspection of the alignment and are shown between tree tips and taxon labels. Functional group assignment is based on Beta-lactamase database (BLDB) and a review." (line 436)

3) Identification of SE

I would suggest combining information on Lines 116-129 and 133-40 and placing after Lines 91-106, first describing the SE identified here, before going on to talk about the patterns.

Lines 95-8 - I would suggest making it clear here that Refs 13 and 15 are your previous work and this part was essentially to finalize analysis of 04Ya311.

(Response) In the revised manuscript, sentences were rephrased as: "Since we previously identified pAQU1 by exogenous plasmid capture using mating and sequencing (13), it is unknown whether the original pAQU1 host strain 04Ya311 carried another copy of *bla*_{GMA-1} and SE in the chromosome, as found in two *V. alfacensis* strains isolated from the same area in Japan as 04Ya311 (15).. To finalize the analysis of 04Ya311,....." Now, we think that it is clear that our previous researches and this study are tightly linked. (line 109)

Lines 102-6 - if there are papers associated with the accession numbers they should be cited, at least in Table S1.

(Response) We added a new column “associated literature” in Table S1, and added relevant references available.

Lines 121-7 - the nucleotide sequences of various SE appear highly related, so I would suggest referring to these rather than “gene product sequences” here - if the nucleotide sequences are highly related it follows that the protein products will be too.

(Response) As reviewer 1 pointed out, the nucleotide sequences of three SEs (SE-Pda04Ya311, SE-VpaV493, and SE-VscVS12C) are very similar. In this revision, we realized that the differences in the protein ID of SE-VscVS12C and SE-Pda04Ya311 core genes were due to annotation error for SE-VscVS12C. Therefore, in the revised manuscript, we handled previous SE-VscVS12C as SE-Pda04Ya311. We also found mistakes in the descriptions of SE termini of these SEs in Table S1. We show the correct SE termini and 6-bp footprints in Table S1. In the revised manuscript, we stated that “SE-VpaV493 is very similar to SE-Pda04Ya311 at nucleotide level. They possess identical terminal sequences (Fig 3A(ii) and Fig 3C(ii)) and a gene encoding DUF262-domain containing protein”. (line 149)

Line 123 - what are these cargo genes and where are they in relation to blaGMA-1? This might be most easily shown on a figure (see comments on Figure 2 below).

(Response) Cargo genes encode hypothetical proteins for both SE-Pda04Ya311 (one large pentagon) and SE-VpaV493 (small three pentagons). We stated that “cargo genes encoding hypothetical proteins,” in line 153. We decided not to add the text to the Figure because other SEs also have hypothetical protein genes, and adding texts would increase its complexity.

pSEA-1 Table S1 lists NZ_LC081338.1 as a duplicate of NZ_AP024167.1, with both plasmids named as pSEA1, but the sequences are slightly different lengths. Are both correct? If not, maybe the incorrect one could be removed from GenBank to reduce confusion.

(Response) The sequence of NZ_LC081338.1 (pSEA1 captured in *E. coli*) was previously determined using only 454 reads many years ago (the original reads were lost when the HDD of server was broken before submitting to SRA); thus, we speculated that the sequence of NZ_LC081338.1 might be incorrect. On the contrary, we also think that NZ_LC081338.1 might be a correct sequence of pSEA1-variant that emerged after rearrangement in the *E. coli* transconjugant. Because the accession number of LC081338.1 is already linked to a published paper, we decided not to delete LC081338.1.

Line 125 - are the SE names correct here? Table S1 and elsewhere in the text have VpaV493, not VdaVS492, and SE-VcVS12C, not SE-VcVS12B.

(Response) We apologize for the incorrect SE names. VpaV493 was correct, but “VdaVS492 and SE-VcVS12C, not SE-VcVS12B” were all incorrect and removed from the revised manuscript.

Fig. 2 - I would suggest changing this to better show the relationships between all SE analysed here and incorporate the information on Lines 133-40. Diagrams are not really needed for the sequences without SE - the accessions could just be stated. Diagrams for the previously-characterized SE also analysed here could be included, with all diagrams aligned and SE names added. Variants of the intA, tfp intB and srp genes could each be shown e.g. in different shades of the same colour, as the different WP_ accessions are not easy to distinguish at a glance. The att site alignments for the new SE could be shown separately.

(Response) We appreciate the suggestion of reviewer 1. SE-6945 is shown in new Fig. 2 as an example of SE for introduction. In the new Fig. 3 (previous Fig. 2), we added a diagram of SE-Pda04Ya311 (panels A) to discuss the variations in 6-bp footprint and integration locations of SE-Pda04Ya311. Since the previous SE-VscVS12C was identical to SE-04Ya311, it is now shown as SE-Pda04Ya311 in panel B. The

accession number information was reduced since it is described in Table S1. Variations in the genetic organization of SE members are shown in the new Fig. 3 panel E. This may help readers easily understand SE gene content variations.

We disagree with the following comments of reviewer 1. "Diagrams are not really needed for sequences without SE - the accessions could just be stated" and "Variants of *intA*, *tfp*, *intB*, and *srap* genes could each be shown in different shades of the same color, as different WP_ accessions are not easy to distinguish at a glance. The *att* site alignments for the new SE could be shown separately." It would be helpful for readers to quickly understand how the *att* sites were detected, and that *attB* is indeed not occupied in the genome being compared. Therefore, we decided to leave the diagram of the SE-free genome and show that *att* alignments blew the gene map as before.

Seven colors are already used just for the gene map. We do not wish to use more colors or patterns for this figure. Orthologous SE proteins are shown in the same color, and protein sequence variations are shown by protein ID.

Fig. 3A - I find this format very difficult to interpret and I don't think it is useful to include the non-SE blaGMA-1 contexts here - this figure is not really needed if the suggested changes to Fig. 2 are made and Fig. 3B could be shown separately or maybe with Fig. 2 or Fig. 4? Legend, Line 389 - "similarity" and "identity" have specific meanings in relation to proteins

(Response) We understood reviewer 1's concern. In the revised manuscript, the previous Fig. 3 (2D blastp) was deleted, and a new paragraph was made, focusing on SE-independent blaGMA-1 acquisition and related figures (new Fig. 4 and new Fig. S1). A discussion on SE-associated blaGMA-1 acquisition was made using new Fig. 3. We believe that the current manuscript organization should help readers easily track the contents and messages.

Line 91 - suggest "SE are the main type of mobile element..."

(Response) We have used "the SE is the main type of" in the revised manuscript because language editor suggested so. (line 108)

Lines 93, 172 - not clear what is meant by "primal" here?

(Response) We have used "the main type of" for both sentences.

Line 98 - suggest "to investigate this"

(Response) "To finalize the analysis of 04Ya311" was stated in the revised manuscript. (line 113)

Line 100 - this is unclear - the only copy is on pAQU1?

(Response) We used "... have two chromosomes, pAQU1, and the only copy of SE-Pda04Ya311 on pAQU1." (line 115)

Line 103 - suggest ">5 kb on either side of blaGMA-1" if this is what is meant?

(Response) We thank reviewer 1 for this comment. We used ">10 kb on either side of *bla*_{GMA-1}." (line 120)

Lines 103-4 - "13 entries including pAQU1" might work better.

(Response) We used "21 filtered RefSeq/GenBank entries, including pAQU1" as suggested. (line 121)

Lines 104-6 - if all species are listed then "included" should be removed.

(Response) We thank reviewer 1 for this remark. We used "are" instead of "included." (line 122)

4) bla_{GMA-1} contexts

I would suggest combining information on Lines 107-116, 130-33 and 141-165 to describe the different "patterns". This information is currently presented in a way that I think is overly complicated and confusing. From Table S1, "patterns" 1-6 just seem to correspond to different SE and this could be made much clearer in the text once all of the SE have been described

(Response) In the revised manuscript, we emphasized that "pattern" refers to the pattern of bla_{GMA-1} acquisition event on the chromosomes or plasmids, which is a unique combination of mobile elements containing bla_{GMA-1} and target location (chromosome/plasmid) (Fig. 3), or an integration of difficult-to-define DNA segments containing bla_{GMA-1} into plasmids or chromosome (Fig. S1). The patterns are not simply bla_{GMA-1} locations.

In the revised Fig. 5, bla_{GMA-1} insertion locations in the same SEs are handled as one location, and those are described as "pattern_A/B" for bla_{GMA-1} in SE-6945 and "pattern_C/D/F" for bla_{GMA-1} in SE-Pda04Ya311 and SE-VpaV493. In the figure legend, we explicitly stated that "Nucleotide sequences of bla_{GMA-1}-flanking regions at nine bla_{GMA-1} insertion locations. The locations are associated with bla_{GMA-1} acquisition patterns. Pattern_A/B indicates the location within SE-6945, whereas pattern_C/D/F indicates the location within SE-Pda04Ya311 and SE-VpaV493." (line 742)

Lines 109-111 - these are all identical SE-6945, so it is not clear why they are considered two "patterns"?
(response) In this manuscript, we introduced the term "pattern" in "bla_{GMA-1} acquisition pattern" in line 140, so cases where SE-6945 was integrated into different locations should be handled as different acquisition patterns. We think that this usage is valid since the focus of this study is on how bla_{GMA-1} moves between genomic locations.

Lines 109-112 - are the whole plasmids and chromosome closely related or just the bla_{GMA-1} contexts?
(response) We deleted the previous sentences. In the revised manuscript, we put more emphasis on variations in SE-Pda04Ya311 insertion locations and its 6-bp footprint using more words and new Fig. 2.

Lines 112-6 - similarly these are all SE-Pda04Ya311 and this could just be stated.
(response) Yes, they are. Currently, we used more words and figure panels to explain these insertion locations. These two are handled as pattern C2 and C1 (new Fig 2A) in the current manuscript.

Lines 118-29 - if this information is moved, as suggested above, patterns 4-6 could just be equated with the relevant SE. Also consider if the "patterns" are really different between highly related SE with some different segments?
(response) We understood the reviewer's confusion. The target locations of highly related SE with some different segments are indeed identical (see *rraB* in new Fig2). "Patterns" refer to the patterns of bla_{GMA-1} acquisition events on chromosomes or plasmids. Therefore, previous patterns 4-6 (current C, D, and E) are handled as different "patterns." We think that a more detailed description about SE insertion locations in the current manuscript should help the readers understand our observations (lines 125-158).

Lines 130-33 - see comments on Fig. 3 above.
(response) These sentences were deleted due to major changes in manuscript organization.

Lines 141-55 - I think that this analysis has some problems. It is not clear why integron components might be expected and the fact that IS and Tn were searched for should also be mentioned before describing the elements that were found.

I am not sure that the presence of any of these transposases is actually relevant - the whole IS elements need to be identified and checked for TSD etc, to see if they are just simple insertions. A quick look suggests that some are not close to blaGMA-1 and maybe associated with different resistance genes. Are the IS actually ISShrf9, ISVsa5 and ISKpn13 or relatives (cut off is 95% identical nt sequences in ISFinder). More information about the types of plasmids and any relationships between them should also be included.

(response) In the revised manuscript, comparative genomics parts were split into SE-mediated blaGMA-1 acquisition (line 125-) and SE-independent blaGMA-1 acquisition (line 165). In the later part, we described the identification of insertion regions by comparative genomics using the new Fig. S1; then, a tyrosine recombinase or IS elements were stated, if relevant. Since we could not find a compelling evidence that IS elements and a tyrosine recombinase are associated with the acquisition of bla by the chromosomes or plasmids, details of ISShrf9, ISVsa5, and ISKpn13-related elements were omitted. One ISShrf9-like element showed a 95% nucleotide identity over the aligned region in blastn against the ISfinder database; one copy showed <95% identity. Therefore, "ISShrf9-like IS" was used for both ISShrf9-like in the revised manuscript (line 189).

Lines 159-63 - it might be good to make it clearer here that one IR is identical in all sequences but the other has differences.

(response) We put an additional sentence: "One inverted repeat (green in Fig 5A) was identical in all sequences, whereas the other had differences in sequence" in the revised manuscript. (line 213)

Fig. 4 - it is not immediately clear what these six sequences represent, without referring back to other figures, or why only these sequences are included - using only the accessions to identify them is not informative.

Could the extra T in the motif in CP007004 be an error? If the reads are available, it would be possible to check this by read mapping.

The legend should explain what * indicate. I would suggest using dots to mark every 10th base rather than numbering all positions.

How were the promoter motifs identified? I couldn't find thsi in methods. The -10 motif seems extended?

(Response) We could not find original reads used to generate the sequence submitted as "CP007004" in SRA. Since SE-VpaV493 (in CP007004) is almost identical to SE-04Ya311, except for the right end, we handled blaGMA-1 locations in SE-04Ya311 and SE-VpaV493 as one (pattern_C/D/F) in the revised manuscript; then, we used the pAQU1 sequence, which does not have an extra "T," as a representative sequence of blaGMA-1 insertion location of pattern_C/D/F. (new Fig 5)

To predict promoter region, a B PROMO server (<http://www.softberry.com/berry.phtml?topic=bprom>) was used, which is not that accurate. In this revision, a SAPPHERE server (<https://sapphire.biw.kuleuven.be>) was used, and promoter motifs were obtained with sufficiently low p-value in a different region. New results of SAPPHERE prediction are shown in the new Fig. 5. We added a statement "Fully conserved nucleotides are indicated by asterisks below the sequences" to the legend.

Line 132 - suggest "six of these nine locations are associated with SE"

(Response) We thank reviewer 1's suggestion. However, the relevant phrase was removed due to major changes in contents.

Line 133 - is there sufficient evidence to generalise beyond the current dataset?

(Response) In this revision, data from both GenBank and RefSeq were included. As of September 2021, we found only 27 hits (including pAQU1) for GMA-1, while 40 GMA-1 hits and 2 GMA-2 hits were found in the new dataset.

In this dataset, “blaGMA-1 alone translocation” cases were newly found in three sequences, one with 5-bp TSD (new Fig. 4). In the revised manuscript, we showed SE-independent blaGMA-1 acquisition patterns in the new Fig. S1. Even after using an increased dataset, SE was the only mobile element repeatedly showing up around *bla*_{GMA-1} and explained eight blaGMA-1 acquisition events, which are more frequent than six SE-independent blaGMA-1 acquisition events observed in the current dataset. Therefore, we think that the conclusion is still valid.

Line 146 - not clear what "respectively" refers to here?

(Response) The sentence was removed from the revised manuscript.

Line 165 - how would this suppress cost?

(Response) We meant to say that the sequences remained unchanged, and gene expression levels are maintained at a harmless level. We rephrased the sentence as “..selection to retain function and an expression level harmless for hosts.” (line 218)

5) Methods

Line 178 - this is one strain? If so "...Japan) was cultured"

(Response) This was corrected as suggested.

Lines 193-7 - primer sequences are usually given in uppercase.

(Response) We updated the primer sequences to uppercase.

Lines 200-2 - how was this determined, i.e. was the recombinant plasmid sequenced? If so, this should be stated.

(Response) We rephrased the sentences as follows: “Plasmids were purified from transformants, and the insert was determined by Sanger sequencing. The recombinant plasmid carrying *bla*_{GMA-1} with the authentic promoter was named pHY1389.” (line 260)

Line 208 - "is not included" "by standard microdilution"

(Response) The two points were corrected as suggested.

Line 210 - "on three"

(Response) We used “on three different days” as suggested.

Line 222 - delete "of".

(Response) We rephrased the sentence as follows: “In total, 39 RefSeq and GenBank hits (Sept 19, 2023)” because language editor suggested so.

Line 225, 232 - the listed names are of genes, not proteins.

(Response) We updated line 225 (*intA*, *tfp*, *intB*, *srpA*) and line 233 (*intB*, *tfp*) to *TntA*, *Tfp*, *IntB*, *SrpA*, *IntB*, and *Tfp*.

Line 235 - "the termini of SE were"?

(Response) This was corrected as suggested.

6) Suggested minor wording changes to improve clarity etc in other parts of the manuscript

Lines 12-3 - genomic data presumably revealed the gene, not the protein?

(Response) We replaced "GMA-1" with "*bla*_{GMA-1}."

Lines 14-6 might be better before Lines 12-3.

(Response) We switched the two sentences.

Lines 17,30 - suggest "in sequences in..."

(Response) For previous line 17, we used "...how frequently *bla*_{GMA-1} is acquired by the chromosomes or plasmids via SEs using sequences in publicly available database." We used "in sequences in" in previous line 30 (new line 33) as suggested.

Line 20 - "...2c, narrow spectrum beta-lactamases." Also see comments above on number of "patterns"

(Response) We used "2c, narrow-spectrum beta-lactamases" in the revised abstract. (line 21)

Line 27 - suggest delete "on class A beta-lactamases"

(Response) "on class A beta-lactamases" was deleted.

Line 167 - "demonstrates", again see comments above on number of patterns,

(Response) This was corrected as suggested (line 220). Regarding the interpretation of *bla*_{GMA-1} acquisition patterns, the messages were changed, considering new data.

Lines 169-70 - suggest "the chromosome"

(Response) We added "the" as suggested.

Line 171 - "hitchhiking on"

(Response) We added "on" as suggested. (line 227)

References - Line 279 - the whole *aac* gene names should be in italics. "beta" in several reference titles probably should be changed to the beta symbol.

(Response) We thank the reviewers for pointing these out. We corrected the *aac(6')*-31 and beta.

Line 364 - "except for carbenicillin"

(Response) We added "for" as suggested.

Table S1

Column B - heading should be "strain/plasmid"?

(Response) We changed the heading to "strain/chromosome/contig/plasmid"

Column C - what does "not open to public" mean - not listed in GenBank? Can this information be obtained from an associated paper or is it suggested by the submitter's location?

(Response) The information was collected from GenBank files or linked Bioproject if available. We added this information to the heading.

Column G "franks" should be "flanks".

(Response) This was corrected. (new Table S1)

Reviewer #2 (Comments for the Author):

The manuscript "Mobile class A β -lactamase gene blaGMA-1" by Yano et al. describes the gammaproteobacterial mobile class A B-lactamase (GMA) gene and how it relates to resistance. The authors also characterize this target through a few approaches. The manuscript has good grammatical structure and details an interesting area of antimicrobial resistance research.

Opportunities:

This manuscript is framed around the genomic/gene components of antimicrobial resistance, however the research detailed in the methods section does not reflect this. I found myself looking for specific information on multiple genomes/isolates and could only identify 1 that was sequenced.

In addition, many of the questions answered in the manuscript are reported as defining features that are already known in the introduction. While this confirms the results, there should be a greater scope to support the research.

There are some interesting areas to explore and the overall manuscript could be strengthened with a broader scope.

(Response) We are grateful for reviewer 2's comments.

(1) Regarding descriptions in the method

To increase the data set, we accessed the NCBI Identical Protein Group website (<https://www.ncbi.nlm.nih.gov/ipg/>) on September 13, 2023. The website returned a table like the example shown in the instruction (<https://www.ncbi.nlm.nih.gov/ipg/docs/about/>), and we copied a total of 38 RefSeq/GenBank IDs that contain information on coding sequences for GMA-1. However, when we tried to do the same thing on October 23, 2023, the website did not return a list as pointed out by reviewer 2.

We think that the current NCBI Identical Protein Group website may have a problem with stably creating a list. Otherwise, the NCBI website may have a problem with compatibility with versions of web-browser or OS. Because of the way the method used is described, we did not add major changes, except for changing "RefSeq" to "RefSeq/GenBank" in the method.

(2) Regarding broad scope contents

To have broad scope contents, additional comparative genomic analysis was conducted by adding GenBank data to RefSeq data. After updating the genome dataset, the observed blaGMA-1 acquisition patterns increased from 11 to 14.

We found three cases of DNA molecules that might have acquired *bla*_{GMA-1} through *bla*_{GMA-1} alone translocation. Notably, 5-bp target site duplication was detected in *V. parahaemolyticus* genome (new Fig. 4).

In the revised manuscript, comparative genomics was split into (1) SE-mediated *bla*_{GMA-1} acquisitions (line 120-159) and (2) SE-independent *bla*_{GMA-1} acquisitions (line 160-198). The latter was emphasized in the previous manuscript.

Even after increasing the dataset, most (8 out of 14) events of *bla*_{GMA-1} acquisition by the chromosomes or plasmids were associated with the SE. Therefore, the conclusion remains that “SE is the main type of mobile DNA element disseminating *bla*_{GMA-1}.” Because *bla*_{GMA-1} translocation was detected in the new dataset, the following statement was added to the abstract”; however, SE is not the only genetic mechanism transmitting *bla*_{GMA-1}.” Since “beta-lactamase gene alone translocation” may also be a new concept in the field of antimicrobial resistance, we believe that the current manuscript content is of interest to a broad audience.

Re: Spectrum02589-23R1 (Mobile class A β -lactamase gene *bla*_{GMA-1})

Dear Dr. Hirokazu Yano:

Your manuscript has been accepted, and I am forwarding it to the ASM production staff for publication. Your paper will first be checked to make sure all elements meet the technical requirements. ASM staff will contact you if anything needs to be revised before copyediting and production can begin. Otherwise, you will be notified when your proofs are ready to be viewed.

Sincerely,
Daria Van Tyne
Editor
Microbiology Spectrum